# Effectiveness of Proprioceptive Body Vibration Rehabilitation on Motor Function and Activities of Daily Living in Stroke Patients with Impaired Sensory Function

**DOI:** 10.3390/healthcare12010035

**Published:** 2023-12-23

**Authors:** Hyunsik Yoon, Chanhee Park

**Affiliations:** 1Chungnam National University Hospital, Daejeon 35015, Republic of Korea; yhs8282@hanmail.net; 2Department of Physical Therapy, Yonsei University, Wonju 26493, Republic of Korea

**Keywords:** stroke, somatosensory evoked potential, balance, ADL, proprioceptive body vibration training

## Abstract

Stroke patients experience impaired sensory and motor functions, which impact their activities of daily living (ADL). The current study was designed to determine the best neurorehabilitation method to improve clinical outcomes, including the trunk-impairment scale (TIS), Berg balance scale (BBS), Fugl-Meyer assessment (FMA), and modified Barthel index (MBI), in stroke patients with impaired sensory function. Forty-four stroke survivors consistently underwent proprioceptive body vibration rehabilitation training (PBVT) or conventional physical therapy (CPT) for 30 min/session, 5 days a week for 8 weeks. Four clinical outcome variables–the FMA, TIS, BBS, and MBI–were examined pre- and post-intervention. We observed significant differences in the FMA, BBS, and MBI scores between the PBVT and CPT groups. PBVT and CPT showed significant improvements in FMA, BBS, TIS, and MBI scores. However, PVBT elicited more favorable results than CPT in patients with stroke and impaired sensory function. Collectively, this study provides the first clinical evidence of optimal neurorehabilitation in stroke patients with impaired sensory function.

## 1. Introduction

The human brain is constantly changing throughout life, and neurological conditions, such as stroke, are especially sensitive to this type of neuroplasticity [1,2,3]. Hemiparesis of the contralateral upper limb is the most prevalent impairment following stroke, affecting over 80% of patients with stroke immediately and over 40% chronically [4]. Stroke is a common impairment of sensory and motor function and activities of daily living (ADL) following a cerebrovascular accident. Additionally, stroke rehabilitation is a development that helps stroke victims with disabilities return to their regular lives and resume activities of daily living through a motor learning process [1,3]. The human brain can continue reorganizing in response to interventions that affect motor function recovery years after an initial stroke impairment [5]. Following a stroke, deficits in somatic sensations, such as touch, warmth, pain, and proprioception, are frequent, with estimated prevalence rates as high as 11–85% [2,3]. Abnormal synergy, including dystonia, spasticity, rigidity, aberrant muscular stress, and muscle weakness, are common problems that accompany a stroke. These impairments impair gait and balance, in addition to restricting daily activities. Functionally, motor issues arising from sensory deficits post-stroke can be summed up as follows: (1) reduced sensory information detection; (2) disrupted execution of somatosensory motor tasks; and (3) limited extremity rehabilitation [6]. Muscle contraction or weakness, alterations in joint laxity and muscle tone, and poor motor control are typical signs of motor impairment [7]. Disabilities in routine tasks, like reaching, picking up, and holding objects, are caused by these impairments. Motor impairments could be associated with additional neurological symptoms that impede the restoration of motor function and require targeted physical therapeutic intervention. Post-stroke deficits in ADL, including personal hygiene, bathing, feeding, toilet use, stair climbing, dressing, and ambulation rates, have been shown to range from 11 to 85% [8].

Conventional physical therapy (CPT) has been widely utilized in patients with stroke to improve sensory and motor functions and ADL, with variable outcomes reported [9]. Additionally, it tries to help patients comprehend their situation better, cope with the challenges brought on by their disability, and avoid secondary consequences. The CPT involved a number of manual interventions through key control points (trunk, pelvis, shoulder, hip, and head). Facilitation and assistance techniques have been used to support muscle activity as well as inhibition to maintain and control movement and posture [10,11]. The CPT was carried out in sitting and standing, supine, and side-lying positions. The focus was on the shoulder and pelvic patterns, as well as their combinations in the rotation trajectory. Chopping, lifting, and rotation of the upper and lower trunk are used in work with the trunk. Stabilization and balance exercises were carried out on a kinesiotherapy couch and rehabilitation ball [12].

Pumprasart et al. demonstrated that CPT could improve tactile sensation (2%) and proprioception (8%) for six weeks in 26 patients with stroke when compared with pretreatment [13]. Vliet et al. reported that CPT improved gross motor function (3%) and ADL (6%) for six months in 120 patients with stroke when compared with movement science therapy [14]. However, the CPT failed to show statistically significant differences in gross motor function, ADL, and sensory function. These inconsistent findings could be attributed to a lack of proprioception and motivation.

Recently, vibration techniques have been employed, in addition to conventional physiotherapy and rehabilitation techniques, as a kind of treatment. We developed proprioceptive body vibration rehabilitation training (PBVT) to enable ADL and motor function in stroke patients with impaired sensory function by providing ample proprioceptive intensity (vibration) for voluntary sensory and muscle movement. PBVT platforms were created and are now often used to improve muscular function in adults and athletes of all ages [15,16]. These platforms may produce mechanical vibrations at various frequencies and magnitudes [17,18]. PBVT provides a systemic vibration stimulus on a platform, with vertical and horizontal oscillatory movement. Patients with stroke have shown significant improvement in muscular function, muscle strength or weakness, and gait function when using PBVT [17,18]. Standing or performing vigorous movements on a vibration platform set on a static surface is the focus of PBVT training. PBVT training was proposed as a possible strategy for enhancing physical functions in earlier research. Additionally, it was proposed that by boosting muscle strength, PBVT enhances muscle function and balance [17]. Several studies have documented that PBVT treatment enhances trunk balance via multiple pathways, such as motor unit activation, modification of the spinal moto neuronal pool’s excitability, and enhanced proprioception [19,20]. When a patient is positioned on the whole-body vibrator’s platform, PBVT therapy produces either a vertical oscillation or horizontal movement. Vibration stimulation is sent from the feet to the whole body through the platform’s contact surface. Consequently, it is anticipated that PBVT therapy will affect postural control by stimulating muscle group Ia and II afferents [21]. Additionally, by identifying muscle stretching and triggering a tonic vibration reflex, PBVT therapy can enhance proprioceptive function. Priplata et al. stated that vibration therapy is an effective way to increase proprioception and can lead to long-lasting postural improvement [22]. A recent study has shown the positive effects of task-oriented training combined with PBVT therapy on certain components of chronic stroke patients’ sitting balance when they are seated [23,24]. Despite the important clinical ramifications of PBVT in stroke with impaired sensory, its beneficial effects on ADL and motor function remain unknown. The primary aim was to ascertain the therapeutic effects of PBVT on ADL and motor function in stroke patients with impaired sensory function. The secondary purpose was to compare the effects of PBVT on trunk stability and static and dynamic balance in stroke patients with impaired sensory.

## 2. Materials and Methods

### 2.1. Participants

Modified Manuscript: Forty-four participants with hemiparetic stroke were recruited from a neurorehabilitation hospital (males 22, females 22; mean age 65.86 ± 8.81 years; post-stroke duration 3 ± 1.86 months; MMSE-K 25.86 ± 2.02). This retrospective study was approved by the Clinical Research Information Service (CRIS) and Institutional Review Board (1041849-201709-BM-093-01). The exclusion criteria were as follows: (1) past history of brainstem or cerebellar stroke; (2) uncontrollably elevated blood pressure (stage 2), measuring more than 160/100 mmHg; (3) cardiopulmonary dysfunctions prohibiting ambulation test; (4) significant and persistent mental illness; (5) bone instability (spinal column instability, non-consolidated fractures, or severe osteoporosis requiring bisphosphonate therapy); (6) additional neurodegenerative conditions, such as Parkinson’s disease and amyotrophic lateral sclerosis; (7) aphasia preventing the ability to communicate discomfort; and (8) normal sensory function (somatosensory evoked potential [SSEP] test P37 < 41.7 ms) [25]. Of the 55 patients recruited, A baseline somatosensory evoked potential (SSEP) test was performed on 44 individuals who satisfied the inclusion criteria, and they were labeled as having abnormal sensations (Table 1).

### 2.2. Clinical Testing Procedure

To remove experimental biases associated with the participants’ expectations, experimental information that could affect the participants was masked until the experiment was completed.

An experimental checklist was used to consistently implement the pretest, intervention, and post-test. The pretest and posttest were performed using standardized clinical outcome measures, including the trunk-impairment scale (TIS), Berg balance scale (BBS), Fugl-Meyer assessment (FMA), and modified Barthel index (MBI), and permission was obtained to use all relevant tools and questionnaires for this study from the appropriate copyright holder(s).

### 2.3. SSEP Test

The Electro Synergy system (Viasys Healthcare; San Diego, CA, USA) was used to measure the central nervous system’s SSEP [11]. In particular, SSEP gauges how an electrical stimulus travels along the posterior tibial nerve. The posterior tibial nerve was stimulated using a bar electrode. The cathode was placed in the middle position between the medial border of the Achilles tendon and the posterior border of the medial malleolus. The cathode stimulated the posterior tibial nerve at 30 mA [25]. SSEP was considered sensory normal if P37 was ˂41.7 ms and sensory impairment if ≥41.7 ms [25]. In total, 44 patients with sensory impairments were included in the study.

### 2.4. Fugl-Meyer Assessment (FMA)

Five domains make up the FMA, a quantitative assessment of motor recovery following a stroke: motor function, sensory function, balance, joint mobility, and joint pain. Since the FMA-LE synergy scale (sub-score II index) assesses volitional or voluntary locomotor movement patterns, including flexor and extensor synergy, it was utilized to assess lower-extremity sensorimotor function and ankle-knee-hip joint function [26]. The maximal hip flexion (abduction/external rotation), maximal knee flexion, and maximal ankle flexion made up the flexor synergistic movement pattern. In contrast, hip extension/adduction, knee extension, and ankle plantarflexion made up the extensor synergistic pattern. Resistance was applied to assure active movement and to evaluate both movement and strength. The ordinal grading scale consisted of values as follows: 0 “cannot perform”, 1 “can partially perform”, and 2 “can completely perform [27]. However, in this study, the lower motor function domain was only examined in order to identify SSEP in the posterior tibial nerve and lower extremity motor recovery function. The FMA has been reported to demonstrate good validity and reliability in patients with stroke (r = 0.88 and r = 0.73, respectively) [28].

### 2.5. Trunk-Impairment Scale (TIS)

The intervention-related changes in dynamic sitting balance, static sitting balance, and selective trunk movements while seated were assessed using the TIS. The static subscale examines: (2) the subject’s ability to sit and keep their legs crossed passively; and (3) the subject’s ability to sit and keep their legs crossed actively. The dynamic subscale encompasses items on lateral flexion of the trunk and unilateral lifting of the hip. To evaluate the coordination of the trunk, the patient is asked to rotate the upper or lower part of his or her trunk 6 times, originating the movements either from the shoulder joint or from the pelvic girdle, respectively. An ordinal scale, a 2-, 3-, or 4-point ordinal scale is utilized for each item. The maximum scores that can be obtained on the static and dynamic seated balance and coordination subscales are 7, 10, and 6 points. The 17 items in the scoring system, which consists of 3 subscales, have a possible score ranging from 0 to 23. This scale has excellent reliability for stroke (r = 0.99) [29].

### 2.6. Berg Balance Scale (BBS)

Patients with balance impairment and stroke were measured using the Balance-Oriented Balance Scale (BBS). The exam consists of fourteen balance-related tasks, which range from standing on one foot to standing up from a seated position. Each task was assigned a score ranging from 0 (unable to perform) to 4 (able to perform independently); the scores were added together to determine the final [14]. This scale has been shown to exhibit excellent intra-rater reliability (0.98) [30].

### 2.7. Modified Barthel Index (MBI)

The MBI is a popular assessment instrument that gauges an individual’s dependence on performing functional ADLs. It is rated 0–5 for bathing, personal hygiene, and wheelchair management; 0–10 for feeding, dressing, transferring to the toilet, climbing stairs, bladder control, and bowel control; and 0–15 for chair/bed transfers and ambulation. Each category is rated on a 5-point scale [19]. A number between 0 and 24 denotes complete dependence, 25 and 49 severe dependence, 50 and 74 moderate dependence, 75 and 90 mild dependence, 91 and 99 minimal dependence, and 100 independences. The MBI has demonstrated good validity and reliability in patients with stroke (r = 0.88) [31].

### 2.8. Experimental Procedures

This study aimed to compare the therapeutic effects between the CPT and PBVT groups. Both groups underwent their respective interventions five days a week for eight weeks, and each intervention session lasted 30 min. The specific instructions for the CPT procedure were as follows: (1) A neurodevelopmental treatment (NDT)-certified therapist first normalized a participant’s sternum, ribcage, and thoracolumbar spine alignment in the standing position to allow for normalized diaphragmatic breathing. The therapist inspected each participant’s movements and made any necessary corrections, and the following exercises were performed: (2) lower extremity strengthening exercises (repetitive sit-to-stand exercise; squat exercise; quadriceps setting exercise), (3) weight-bearing exercises on the affected side feet (Forward lunge, Front/Back/Lateral weight shift), (4) static and dynamic standing balance training (Tandem stand, Feet together), (5) visual deprivation standing balance training (Romberg stance-Eye closed), (6) one-leg standing balance training, and (7) stepping exercises (Step up, Side step). The PBVT group stood on a body vibration platform (Novotec Medical GmbH, Pforzheim, Germany) and underwent the same procedures as the CPT group. The amplitude range and frequency of the PBVT intervention were 20–30 Hz [32].

### 2.9. Statistical Analysis

Descriptive statistical data are presented as means and standard deviations. The independent *t*-test was used to compare baseline demographics and clinical characteristics between groups for continuous variables, and the chi-squared test was used for categorical variables. Paired *t*-tests were conducted to examine the differences between the pre- and post-tests for the individual groups. To confirm the post-intervention differences between the groups, independent *t*-tests were run. A 0.05 threshold for statistical significance (α) was established. Statistical analysis was conducted using the Statistical Package for the Social Sciences (SPSS) for Windows version 26.0 (SPSS, Chicago, IL, USA).

## 3. Results

In the FMA, TIS, BBS, and MBI measurements, both groups showed significant differences between pre- and post-test values (*p* < 0.01). In addition, according to the independent *t*-test of differences in mean pre- and post-test values, comparing the FMA and MBI test values was significantly increased by 10.97%/19.86% in PBVT than CPT after intervention (*p* < 0.05). The BBS test was significantly increased by 29.83% in PBVT than CPT after intervention (*p* < 0.01). comparing the PBVT and CPT groups, and the FMA and MBI test values differed significantly post-intervention (*p* < 0.05). The BBS test results differed significantly between the PBVT and CPT groups (*p* < 0.01). However, there was no significant difference in the TIS outcome measurements between the PBVT and CPT groups (Table 2).

## 4. Discussion

To the best of our knowledge, this is the clinical intervention on whole-body vibration to evaluate the effects of proprioceptive body vibration training (PBVT) and conventional physical therapy (CPT) on the trunk-impairment scale (TIS), Berg balance scale (BBS), Fugl-Meyer assessment (FMA), and modified Barthel index (MBI) associated with stroke in stroke patients with impaired sensory function. Our hypothesis was supported by the fact that, in terms of voluntary movement, abnormal synergy, static, and dynamic balance, and ADLs, the PBVT group significantly outperformed the CPT group. Importantly, the PBVT group exhibited clinically more meaningful improvements in voluntary movement, abnormal synergy, static and dynamic balance, and ADL in stroke patients with impaired sensory impairment than the CPT group.

Furthermore, the PBVT group showed a marked improvement in the FMA score (78.3%) when compared with the CPT group (36.5%). These findings corroborate those of earlier research looking at the aberrant synergistic therapeutic effects of PBVT in stroke patients experiencing sensory impairment. Cordo et al. documented improved abnormal synergy after 30 sessions of muscle vibration (16.8%) when compared with electromyography-biofeedback intervention in 32 participants with severe hand impairment due to chronic stroke [33].

In 29 participants with chronic stroke, Lee et al. reported an increased FMA score after 14 min of whole-body vibration (0.18%) when compared with the control group intervention [18].

BBS analysis also demonstrated a superior positive effect of PBVT (31.34%) over CPT, which is parallel with the results of previous studies that examined the therapeutic effects of PBVT on dynamic and static balance in patients with stroke. Marin et al. reported improved balance, muscle strength, and muscle architecture scores after 17 sessions of whole-body vibration (27%) compared with those in the sham group in 11 patients with stroke [34]. Chuan et al. reported that dynamic balance, including walking performance, improved following 8 weeks of whole-body vibration training (14%) compared with the control group in 30 patients with stroke [35]. PBVT training has been used extensively to support stroke patients’ rehabilitation, and some evidence of its efficacy in recovering dysfunction in neurological diseases has been shown [36]. The whole body is stimulated by the impact stimulation generated by PBVT training, which also increases the excitability of sensory nerve fiber endings, stimulates proprioception receptors in the form of muscle and tendon spindles, and causes the skeletal muscle to stretch reflexively [37]. As a result, there is an increase in the level of muscle activation and latent motor unit activation. Muscles therefore recruit more motor units [38]. Enhancing muscle strength and improving athletic performance both benefit from increased muscle recruitment efficiency. According to a number of studies, PBVT training can improve muscle activation, balance ability, strength, flexibility, burst force, and electromyography (EMG) signals in the lower limbs of older people [39,40]. As a result, when walking, the body’s center of gravity moved in a wide range of amplitudes from left to right. More autonomic control is needed on the unaffected side to ensure adequate stability.

In the current study, PBVT elicited a superior improvement in the ADL analysis (61.3%) when compared with CPT (26.8%) in 44 stroke patients with impaired sensory function. This finding was consistent with the results of Hwang, who reported an improved MBI score after four weeks of whole-body vibration (154%) when compared with the control intervention (46.6%) in 18 participants with subacute stroke [41]. Van Nes et al. found that 6 weeks of whole-body vibration increased ADLs (103.7%) more than sham treatment (100%) [42]. In 34 patients with stroke, Teresa et al. reported superior improvement with vibration therapy (frequency of 50 Hz and amplitude of 2 mm; 70.6%) when compared with that observed in the control group (41.2%) [43].

It could be postulated that the selected PBVT intensity and duration were markedly low to encourage lasting changes in the somatosensory pathways or the sensorimotor cortex. However, we selected intensity and duration comparable to those used in previous studies in healthy subjects and nursing home residents who had reported the beneficial effects of PBVT. We adjudged that a more robust intensity than that selected is unwarranted in patients with stroke, particularly because PBVT has been shown to induce early muscle fatigue compared with regular muscle exercises [44]. Even in healthy individuals, muscle fatigue occurs after a few minutes of stimulation [45].

However, TIS analysis didn’t effectively improve trunk stability function compared to CPT. While Lau and colleagues reported that PBVT had no effect on falling or gross motor functions, their study focused on self-efficacy of falls, which is associated with dynamic and static balance, trunk movement control, mobility, and muscle strength [46]. Although it cannot be definitively concluded that PBVT directly improves trunk motor functions and thereby prevents falls, PBVT may undoubtedly help people who experienced strokes from developing subsequent issues brought on by an unintentional fall risk [46]. In terms of muscular strength and balance, Tihanyi and colleagues found that PBVT was successful in raising voluntary muscle strength, which improved trunk control and balance even more [47]. Lau and colleagues reported that PBVT improved trunk functioning and reduced the chance of falls in stroke patients [46]. After a 12-week training, van Nes and colleagues found that the PBVT group outperformed the controls in terms of balance and performance of daily activities [42]. Choi and colleagues recommended PBVT as a useful training technique to enhance balance and improve seated balance [48]. Additionally, postural sway augmentation, posture correction, and vestibular system stimulation were all mentioned as benefits of PBVT [49]. Similarly, Yule and colleagues found that while PBVT is a safe means to improve abnormal spasticity, which is associated with safety and sitting static balance, it does not effectively increase physical functions associated with muscle strength and balance that are linked to walking and activities of daily life [50]. Likewise, Liao and colleagues and Tankisheva colleagues reported that PBVT is a safe treatment therapy that can be used to enhance physical functions or structure, activity, and muscle gain [51,52]. Because PBVT affects bone mineral density and safely lowers spasticity, it may help avoid secondary issues brought on by unintentional falls [50]. There was no variation in the size of the impacts based on frequency, despite the fact that these studies compared the effects of vibration frequencies [50]. Mileva and colleagues found that this enhancement is due to an increase in excitability of the corticospinal pathways, which explains why cortical structures are involved in the acute effects of vibration [21]. On the other hand, intense peripheral stimulation and the more effective application of vibration-induced proprioceptive feedback could be the source of the improvement [21]. Previous studies have shown that when vibration therapy methods are delivered directly to tendons, factors including frequency, amplitude, and tendon loading significantly impact the sensitivity of the muscle spindle afferents. According to Kihlberg et al., PBVT at a frequency of 50 Hz may be more effective at inducing muscle activation compared with a higher frequency of 137 Hz [53]. Mester et al. recommended not performing vibration training with frequencies lower than 20 Hz to avoid the body resonance frequency range with its damaging effects [54]. Cardinale and Lim demonstrated the electromyography (EMG) activity of the vastus lateralis muscle under PBVT at frequencies of 30, 40, and 50 Hz in individuals without neurologic disorders and found a significant increase at a frequency of 30 Hz, which denotes better synchronization of the motor units [55].

The Galileo equipment used in the current study was a side-alternating white blood cell device. Side-alternating vibrations present three characteristics. First, the anteroposterior mediolateral axis was the center of rotation for the platform, and stronger amplitude vibrations were produced when the axis of rotation was farther from the point of application [56]. Second, by inducing rotational movements around the hip and lumbosacral joints, the device gives the therapy an extra dimension of flexibility. Consequently, the side-alternating PBVT presents a lower whole-body mechanical impedance than the synchronous PBVT and can be expected to reduce vibration transmission to the trunk and head [25,26]. Finally, lower limb muscles exhibit significantly greater electromyography activity during side-alternating vibration compared to synchronous vibration. The amalgamation of high vibration frequencies at 30 Hz and extra weight placed on a platform with side-alternating vibration has been reported to be associated with maximum EMG activity during PBVT exposure. Therefore, it is speculated that side-alternating whole-body vibration is the most efficient method for PBVT training [57]. Tihanyi et al. showed that the PBVT group experienced a substantially greater increase in isometric knee extension torque and eccentric knee extension torque on the paretic side compared with the control group, which performed the same exercises without PBVT [47]. Simultaneous coactivation of the antagonist and agonist muscles led to improvements in balance and daily living movements in the PBVT group [47]. Presynaptic terminals may experience the depletion of neurotransmitters or the presynaptic inhibition of Ia afferents as a result of the vibration stimulus. This may limit the stretch and H re-flexes of the muscles, lessen the excitability of the monosynaptic reflex, lessen abnormal spinal reflex excitation, and control muscle spasm [58,59]. Hence, the balance, abnormal synergistic movement, and ADL function of patients with stroke were improved [38,60]. Studies have shown that following PBVT training in stroke patients, the amplitudes of the soleus and gastrocnemius H reflexes in the lower leg were significantly decreased [61,62,63]. Other vibration effects on spasticity, including presynaptic inhibition and post-activation depression, should be considered. H-reflex amplitudes are inhibited when the presynaptic inhibition of Ia-afferents diminishes the effects of Ia-afferents on motoneurons by reducing the release of neurotransmitters to the motoneurons. Comparing PBVT training to other rehabilitation training methods, its main advantage is that it is a relatively passive exogenous stimulus and can achieve effective training with a smaller load intervention, which lowers the cardiopulmonary burden. The potential benefit of PBVT is enhanced dynamic and static balance, abnormal synergistic movement, and ADL such as dressing, washing, and bathing in stroke patients with impaired sensory.

Several limitations of the current study should be addressed in future studies. First, the sample size of both groups was small, given the difficulties in recruiting participants due to the risk of falling. Therefore, clinicians should cautiously interpret these results, although the clinical outcome measurement data demonstrated promising PBVT control effects. SSEPs primarily assess the integrity of the dorsal (sensory) pathways of the spinal cord. In this study, the SSEP test was used to categorize patients with sensory impairments. However, we were unable to evaluate the patient’s sensory abilities after treatment because our objective was to assess motor skills and ADL. In our next study, we will add sensory abilities to evaluate stroke patients. Further large-scale studies should be conducted to confirm the positive findings of PBVT on motor function, synergistic movement, and balance following stroke. Second, this study was in the absence of a gender stratification and stroke side, which warrants the need for stroke affected side and gender difference in muscle characteristics. Lastly, the current research was not a double-blind study. However, stroke patients with impaired sensory function may benefit from a transdisciplinary and comprehensive approach to rehabilitation that is started in centers that are well equipped with a variety of specialties. This will improve long-term outcomes by helping to diagnose the condition and develop appropriate interventions.

## 5. Conclusions

Herein, our findings revealed that PBVT could afford superior improvements in affected abnormal synergistic movement, motor function, and ADLs than CPT in stroke patients with impaired sensory function. However, PBVT did not impact the TIS scores. Moreover, proprioceptive vibration training rehabilitation is beneficial for gross motor, voluntary movement, and ADL. Our novel results have important clinical implications for stroke neurorehabilitation. Hence, it is crucial to appropriately consider PBVT, which offers comfortable gross motor movements and ADL. Moreover, it facilitates a safer environment than conventional stroke rehabilitation.

## Figures and Tables

**Table 1 healthcare-12-00035-t001:** Demographic and clinical characteristics of the participants (N = 44).

Parameters		^a^ PBVT (*n* = 22)	^b^ CPT (*n* = 22)	*p*-Value
Gender	Male	12	10	0.55
	Female	10	12	
Paretic side	Right	14	11	0.27
	Left	8	11	
Age (years)		64.68 ± ^d^ 8.82	67.04 ± 8.85	0.38
^c^ MMSE-K		26.31 ± 2.00	25.40 ± 1.99	0.13
Post-stroke (months)		1.95 ± 0.65	1.77 ± 0.68	0.37

Note: ^a^ PBVT, Proprioceptive body vibration training; ^b^ CPT, Conventional physical therapy; ^c^ MMSE-K, Korean mini-mental state examination; ^d^ Mean ± Standard deviation.

**Table 2 healthcare-12-00035-t002:** Differences in clinical test scores between PBVT and CPT.

		PBVT (n = 22)	CPT (n = 22)	*p*-Value
FMA (score)	before	12.36 ± 4.49	14.54 ± 4.00	
	after	22.04 ± 3.53	19.86 ± 3.53	0.04 *
	t	11.58 **	11.62 **	
TIS (score)	before	13.95 ± 5.41	14.27 ± 4.77	
	after	19.54 ± 2.64	18.86 ± 3.38	0.46
	t	5.88 **	5.96 **	
BBS (score)	before	23.90 ± 16.57	15.63 ± 11.94	
	after	40.95 ± 12.24	31.54 ± 10.90	0.01 **
	t	6.65 **	7.95 **	
MBI (score)	before	43.40 ± 19.22	46.04 ± 17.14	
	after	70.00 ± 18.19	58.40 ± 16.32	0.03 *
	t	10.89 **	5.06 **	

PBVT: Proprioceptive body vibration training; CPT: Conventional physical therapy; * *p* < 0.05, ** *p* < 0.01; FMA: Fugl-Meyer Assessment; TIS: Trunk Impairment Scale; BBS: Berg Balance Scale; MBI: Modified Bathel Index.

## Data Availability

The data presented in this study are available on request from the corresponding author.

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
