# Peer review of "Effectiveness of Proprioceptive Body Vibration Rehabilitation on Motor Function and Activities of Daily Living in Stroke Patients with Impaired Sensory Function"

_healthcare, 2023, doi:10.3390/healthcare12010035_

Round 1

Reviewer 1 Report

Comments and Suggestions for Authors

The manuscript is well-structured, with clear objectives, reliable methods, and thorough analysis of the results. These findings provide valuable information for the field of stroke rehabilitation. Despite its limitations, the study presents strong evidence supporting the effectiveness of Proprioceptive Body Vibration Therapy (PBVT) over Conventional Physical Therapy (CPT) in improving certain motor functions and daily activities in stroke patients with impaired sensory function. It is recommended to accept the manuscript with minor revisions

Major Issues:

Section 2 - Description of Participants:"The description of participants in Section 2 requires further detail, including their baseline characteristics. This will help readers better understand the representativeness of the sample and the applicability of the study results."

Section 2.8 - Detailed Description of PBVT and CPT Procedures:"In Section 2.8, a more detailed description of the specific operational steps of PBVT and CPT is needed.

Section 3 - Analysis and Explanation of TIS Scores:"For the non-significant improvement in TIS scores mentioned in Section 3, a more in-depth analysis and explanation are required."

Section 4 - Discussion on Limitations:"The discussion in Section 4 needs to delve deeper into the impact of the study's limitations, such as the small sample size, and discuss how these can be addressed in future research."

Minor Issues:

Section 3 (Results) - Clarification of Group Comparisons:"In Section 3, the description comparing FMA, BBS, and MBI scores between the PBVT and CPT groups could lead to confusion. The text states 'significant differences were observed between the PBVT and CPT groups in FMA, BBS, and MBI scores,' but does not specify the nature of these differences, such as which group performed better or the exact values of the differences. A clearer description of the differences between the two groups is necessary, such as: 'In terms of FMA, BBS, and MBI scores, the PBVT group showed more significant improvements compared to the CPT group, with specific values being... (provide specific values).'"

Section 4 (Discussion) - Organizing the Discussion on Benefits:"In Section 4, the discussion on the potential benefits of PBVT for stroke patients seems scattered across multiple paragraphs without a focused summary. It would be beneficial to reorganize this section to consolidate the discussion on the potential benefits of PBVT into a dedicated subsection, thereby enhancing the organization and readability of the paper."

Comments on the Quality of English Language

"In Section 2, some sentences, particularly those describing the experimental procedures, exhibit overly complex grammatical structures with numerous subordinate clauses and parenthetical elements. This complexity could potentially hinder reader comprehension. It is recommended to simplify these sentences by breaking down complex information into several shorter and clearer sentences."

Lack of Explanatory Text for Graphs and Charts:"Although the graphs and charts in the paper effectively represent data, they may lack sufficient explanatory text to aid readers in understanding the presented data. It would be beneficial to provide brief descriptions beneath the graphs and charts, summarizing their main findings and significance."

Author Response

December 13, 2023

Dr. Jacobo Rodríguez-Sanz

Editor-in-Chief

Healthcare

Re: Effectiveness of Proprioceptive Body Vibration Rehabilitation on Motor Function and Activities of Daily Living in Stroke Pa-tients with Impaired Sensory Function

Dear Dr. Jacobo Rodríguez-Sanz

Thanks for the wonderful opportunity to resubmit our amended manuscript as a "De Novo" manuscript. Please find uploaded the revised manuscript entitled “Effectiveness of Proprioceptive Body Vibration Rehabilitation on Motor Function and Activities of Daily Living in Stroke Pa-tients with Impaired Sensory Function.”  A detailed response to the reviewers’ concerns has also been provided in the authors’ response letter and highlighted in yellow with the specific line numbers in the revised manuscript. 

If you require any further information, please feel free to contact me. We look forward to receiving a favorable consideration.

Sincerely,

Chanhee Park

Comments and Suggestions for Authors

The manuscript is well-structured, with clear objectives, reliable methods, and thorough analysis of the results. These findings provide valuable information for the field of stroke rehabilitation. Despite its limitations, the study presents strong evidence supporting the effectiveness of Proprioceptive Body Vibration Therapy (PBVT) over Conventional Physical Therapy (CPT) in improving certain motor functions and daily activities in stroke patients with impaired sensory function. It is recommended to accept the manuscript with minor revisions:

Major Issues:

Section 2 - Description of Participants:"The description of participants in Section 2 requires further detail, including their baseline characteristics. This will help readers better understand the representativeness of the sample and the applicability of the study results."

Authors response: This was revised (Lines 110-114).

Section 2.8 - Detailed Description of PBVT and CPT Procedures:"In Section 2.8, a more detailed description of the specific operational steps of PBVT and CPT is needed.

Authors response: This was revised (Lines 200-212).

Section 3 - Analysis and Explanation of TIS Scores:"For the non-significant improvement in TIS scores mentioned in Section 3, a more in-depth analysis and explanation are required."

Authors response: This was revised (Lines 317-330).

Section 4 - Discussion on Limitations:"The discussion in Section 4 needs to delve deeper into the impact of the study's limitations, such as the small sample size, and discuss how these can be addressed in future research.

Authors response: This was revised (Lines 394-404).

Minor Issues:

Section 3 (Results) - Clarification of Group Comparisons:"In Section 3, the description comparing FMA, BBS, and MBI scores between the PBVT and CPT groups could lead to confusion. The text states 'significant differences were observed between the PBVT and CPT groups in FMA, BBS, and MBI scores,' but does not specify the nature of these differences, such as which group performed better or the exact values of the differences. A clearer description of the differences between the two groups is necessary, such as: 'In terms of FMA, BBS, and MBI scores, the PBVT group showed more significant improvements compared to the CPT group, with specific values being... (provide specific values).'"

Authors response: This was revised (Lines 248-256).

Section 4 (Discussion) - Organizing the Discussion on Benefits:"In Section 4, the discussion on the potential benefits of PBVT for stroke patients seems scattered across multiple paragraphs without a focused summary. It would be beneficial to reorganize this section to consolidate the discussion on the potential benefits of PBVT into a dedicated subsection, thereby enhancing the organization and readability of the paper."

Authors response: This was revised (Lines 389-391).

"In Section 2, some sentences, particularly those describing the experimental procedures, exhibit overly complex grammatical structures with numerous subordinate clauses and parenthetical elements. This complexity could potentially hinder reader comprehension. It is recommended to simplify these sentences by breaking down complex information into several shorter and clearer sentences."

Authors response: This was revised (Lines 200-212).

Lack of Explanatory Text for Graphs and Charts:"Although the graphs and charts in the paper effectively represent data, they may lack sufficient explanatory text to aid readers in understanding the presented data. It would be beneficial to provide brief descriptions beneath the graphs and charts, summarizing their main findings and significance."

Authors response: This was revised (Tables 1&2).

Reviewer 2 Report

Comments and Suggestions for Authors

Introduction

-Please specify what is Conventional Physical Therapy. You cited proprioception excercise but also Bobath...but in the method section you describe total body excercise as Conventional Physical Therapy. 

- Add more references that support your introduction

- the primary aim and the secondary aim should be clarified.

Method

-Please add the number of istitutional review bord.

- When the study was conducted?

- it is an observational study?

- a gender and side stratification should be used

- it would be interesting to add a control group with a combination with PBVT and CPT

Discussion

-there are many abbreviation that make the text difficult to read, please in the first paragraph of discussion add the entire wording

- the main limitaion of the study is the absence of a gender stratification, please consider the follow article concerning gender differences in muscle characteristics:

Deodato, M., Saponaro, S., Šimunič, B. et al. Sex-based comparison of trunk flexors and extensors functional and contractile characteristics in young gymnasts. Sport Sci Health (2023). https://doi.org/10.1007/s11332-023-01083-7

- the absence of a stratificastion of stroke side is an other important limitation that should be justified.

Author Response

December 13, 2023

Dr. Jacobo Rodríguez-Sanz

Editor-in-Chief

Healthcare

Re: Effectiveness of Proprioceptive Body Vibration Rehabilitation on Motor Function and Activities of Daily Living in Stroke Pa-tients with Impaired Sensory Function

Dear Dr. Jacobo Rodríguez-Sanz

Thanks for the wonderful opportunity to resubmit our amended manuscript as a "De Novo" manuscript. Please find uploaded the revised manuscript entitled “Effectiveness of Proprioceptive Body Vibration Rehabilitation on Motor Function and Activities of Daily Living in Stroke Pa-tients with Impaired Sensory Function.”  A detailed response to the reviewers’ concerns has also been provided in the authors’ response letter and highlighted in yellow with the specific line numbers in the revised manuscript. 

If you require any further information, please feel free to contact me. We look forward to receiving a favorable consideration.

Sincerely,

Chanhee Park

Comments and Suggestions for Authors

Introduction

-Please specify what is Conventional Physical Therapy. You cited proprioception excercise but also Bobath...but in the method section you describe total body excercise as Conventional Physical Therapy. 

Authors response: This was revised (Lines 53-61/200-212).

- Add more references that support your introduction

Authors response: This added the references.

- the primary aim and the secondary aim should be clarified.

Authors response: This was revised (Lines 97-100).

Method

-Please add the number of istitutional review bord.

Authors response: This was revised (Lines 110-114).

- When the study was conducted?

Authors response: This study was conducted in March 2023.

- it is an observational study?

Authors response: This article was retrospective study design

- a gender and side stratification should be used

Authors response: Although we were unable to apply the stratification model in this study, we will perform the effectiveness of rehabilitation based on the grade of sensory impairment in our next study. Thank you for your elaborate comment and we added the limitation of study.

- it would be interesting to add a control group with a combination with PBVT and CPT

Authors response: The next study will attempt to determine the effectiveness of PBVT and CPT in combination for stroke rehabilitation. Thank you for your elaborate comment.

Discussion

-there are many abbreviation that make the text difficult to read, please in the first paragraph of discussion add the entire wording

Authors response: This was revised (Lines 260-262).

- the main limitaion of the study is the absence of a gender stratification, please consider the follow article concerning gender differences in muscle characteristics:

Authors response: This was revised (Lines 394-404).

- the absence of a stratificastion of stroke side is an other important limitation that should be justified.

Authors response: This was revised (Lines 394-404).

Reviewer 3 Report

Comments and Suggestions for Authors

Recommend identifying limitations of study including small sample size, imaging data missing. Limitations incudes no double blind study.

limitations of sensory deficits testing. 

The conclusion from the data is overreaching to say physical therapy only resulted in improvement in balance.

can restate as most probable 

Author Response

December 13, 2023

Dr. Jacobo Rodríguez-Sanz

Editor-in-Chief

Healthcare

Re: Effectiveness of Proprioceptive Body Vibration Rehabilitation on Motor Function and Activities of Daily Living in Stroke Pa-tients with Impaired Sensory Function

Dear Dr. Jacobo Rodríguez-Sanz

Thanks for the wonderful opportunity to resubmit our amended manuscript as a "De Novo" manuscript. Please find uploaded the revised manuscript entitled “Effectiveness of Proprioceptive Body Vibration Rehabilitation on Motor Function and Activities of Daily Living in Stroke Pa-tients with Impaired Sensory Function.”  A detailed response to the reviewers’ concerns has also been provided in the authors’ response letter and highlighted in yellow with the specific line numbers in the revised manuscript. 

If you require any further information, please feel free to contact me. We look forward to receiving a favorable consideration.

Sincerely,

Chanhee Park

Comments and Suggestions for Authors

Recommend identifying limitations of study including small sample size, imaging data missing. Limitations incudes no double blind study.

Authors response: This was revised (Lines 394-405).

-limitations of sensory deficits testing. 
Authors response: Somatosensory evoked potentials (SSEPs) primarily assess the integrity of the dorsal (sensory) pathways of the spinal cord. In this study, the SSEP test was used to categorize patients with sensory impairment. However, we were unable to evaluate the patient's sensory abilities after treatment because our objective was to assess motor skills and activities of daily living. In our next study, we will add sensory abilities to evaluate stroke patients. Thank you for your elaborate comment. This was revised (Lines 394-400).

The conclusion from the data is overreaching to say physical therapy only resulted in improvement in balance.

 Authors response: This was revised (Conclusion section).

can restate as most probable